# AniFaceGAN: Animatable 3D-Aware Face Image Generation for Video Avatars

**Yue Wu**[1][*]  **Yu Deng**[2][*]  **Jiaolong Yang**[3][†]  **Fangyun Wei**[3]  **Qifeng Chen**[1]  **Xin Tong**[3]

[1]HKUST  [2]Tsinghua University  [3]Microsoft Research

## Abstract

Although 2D generative models have made great progress in face image generation and animation, they often suffer from undesirable artifacts such as 3D inconsistency when rendering images from different camera viewpoints. This prevents them from synthesizing video animations indistinguishable from real ones. Recently, 3D-aware GANs extend 2D GANs for explicit disentanglement of camera pose by leveraging 3D scene representations. These methods can well preserve the 3D consistency of the generated images across different views, yet they cannot achieve fine-grained control over other attributes, among which facial expression control is arguably the most useful and desirable for face animation. In this paper, we propose an animatable 3D-aware GAN for multiview consistent face animation generation. The key idea is to decompose the 3D representation of the 3D-aware GAN into a template field and a deformation field, where the former represents different identities with a canonical expression, and the latter characterizes expression variations of each identity. To achieve meaningful control over facial expressions via deformation, we propose a 3D-level imitative learning scheme between the generator and a parametric 3D face model during adversarial training of the 3D-aware GAN. This helps our method achieve high-quality animatable face image generation with strong visual 3D consistency, even though trained with only unstructured 2D images. Extensive experiments demonstrate our superior performance over prior works. Project page: `https://yuewuhkust.github.io/AniFaceGAN/`

## 1  Introduction

Face image synthesis and animation have been a longstanding task in computer vision and computer graphics with a wide range of applications such as virtual avatars and video conferencing. Remarkable progress has been achieved in recent years with a large volume of methods proposed [50, 2, 19, 57, 52, 69, 58, 56, 63, 1, 67, 70, 46, 77]. This progress is hinged on a number of advances in machine learning within which generative adversarial networks (GANs) are arguably the core underpinning.

However, most existing face GANs are based on 2D convolutional neural networks (CNNs) and do not model the underlying 3D facial geometry. When synthesizing faces under different poses and expressions, the results cannot maintain strict 3D consistency. Consequently, these methods can be used in interactive face manipulation but are not suitable for high-quality face video generation and animation. To alleviate this problem, some methods incorporate 3D priors into the generation process [11, 64, 30, 20, 51] for better 3D rigging over the 2D images. Perhaps the most relevant work to ours is DiscoFaceGAN [11], which considers an unconditional and disentangled generative modeling setup as we do. Still, exact 3D consistency cannot be guaranteed by the aforementioned

---

[*]Work done when YW and YD were interns at Microsoft Research.

[†]JY is the corresponding author.

36th Conference on Neural Information Processing Systems (NeurIPS 2022).

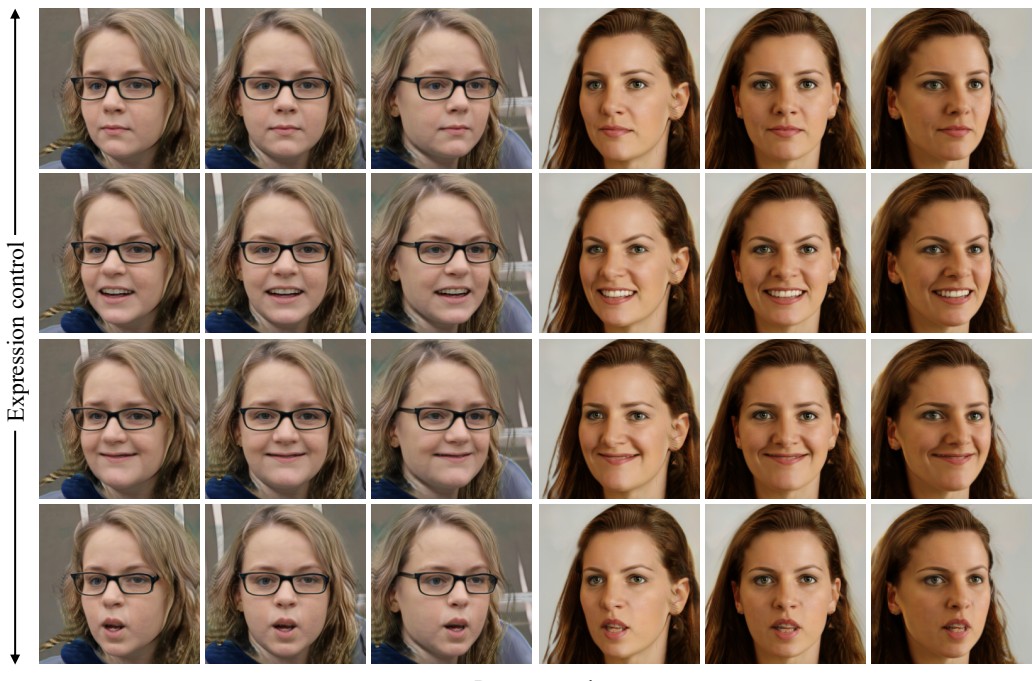

Expression control

Pose control

Figure 1: Random virtual persons generated by our method under different expressions and head poses. Note the high texture consistency across poses and expressions (see more animated examples in our accompanying video).

approaches, and their 2D CNN generators often lead to temporal artifacts such as flickering and texture sticking [25], which are undesirable for realistic video avatars.

Recently, a number of 3D-aware GANs are proposed by incorporating 3D representations [37, 31, 38, 10, 14, 22, 12, 9, 41] to achieve disentangled control of 3D pose. Among them, methods that generate neural radiance fields (NeRF) [35] have demonstrated striking image synthesis results [10, 22, 12, 9]. Owning to its 3D field modeling and volumetric rendering scheme, the NeRF representation is capable of producing realistic images with strong 3D consistency across different views, rendering it suitable for high-quality 3D scene synthesis. Nevertheless, these methods lack the control over attributes beyond camera pose and thus cannot be directly applied to face animation synthesis tasks.

In this paper, we present *AniFaceGAN*, an animatable 3D-aware face generation method that is used to synthesize realistic face images with controllable pose and expression sequences. Our method is an unconditional generative model that can generate novel, non-existing identities and is trained on unstructured 2D face image collections without any 3D or multiview data. To achieve animation, we leverage separate latent representations for identity and expression in the generator, and attain explicit controllability by incorporating priors of a 3D parametric face model. To ensure geometry and texture consistency under expression change, we leverage 3D deformation to derive the desired expressions. Although deformation has been used in some recent NeRF methods [53, 44, 16, 45], these works mostly focused on modeling single dynamic scenes from videos. How to learn deformation in a generative setting from unstructured 2D images and how to achieve explicit and accurate expression control through deformation with unsupervised learning remains underexplored.

Our AniFaceGAN generates two 3D fields for face image rendering: a template radiance field for modeling the geometry and appearance of a generated identity and an expression-driven deformation field for animation. The former is based on the recent generative radiance manifold (GRAM) approach that shows state-of-the-art 3D-aware image generation quality [12]. To learn desired deformation, we incorporate a 3D face morphable model (3DMM) [5, 47] into adversarial training and enforce our deformation field to imitate it under expression variations. In contrast to previous methods such as [11] which imposes imitations on 2D images, we propose a set of 3D-space losses defined on both facial geometry and expression deformation. We train our method on the FFHQ dataset [26] and show that it can generate high-quality and 3D consistent images of virtual subjects across different poses and expressions (an example is shown in Fig. 1).

The contributions of our work can be summarized as follows:

- We present an animatable 3D-aware face GAN with expression variation disentangled through our proposed expression-driven 3D deformation field.
- We introduce a set of novel 3D-space imitative losses, which align our expression generation with a prior 3D parametric face model to gain accurate and semantically-meaningful control of facial expression.
- We show that our method can generate realistic face videos with high visual quality. We believe it opens a new door to face video synthesis and photorealistic virtual avatar creation.

## 2 Related work

**Neural implicit scene representations.** Neural implicit functions have been used in numerous works [42, 34, 60, 59, 35, 39] to represent 3D scenes in a continuous domain. Among them, NeRF [35, 3] shows its superiority at modeling complex scenes with detailed appearance and synthesizing multi-view images with strict 3D consistency. The original NeRF and most of its successors [43, 33, 45, 49, 48, 40, 32, 73] focus on learning scene-specific representation using a set of posed images or a video sequence of a static or dynamic scene. A few methods [55, 10, 38, 76, 22, 12, 9] have explored the generative modeling task using unstructured 2D images as training data. A very recent method GRAM [12] constrains radiance field learning on a set of 2D manifolds and shows promising results for high-quality and multi-view consistent image generation. Our method is also built upon the radiance manifolds representation [12] for high-quality face image generation and animation.

**3D-aware generative model for image synthesis.** Generative adversarial networks [21, 26, 27] are widely used for realistic 2D image synthesis. Recent methods [36, 62, 37, 31, 38, 10, 14, 22, 12, 9, 41, 72] extend GANs to 3D-aware image synthesis by incorporating 3D knowledge into their generators. This enables them to learn multiview image generation given only unconstrained 2D image collections as supervisions. For example, HoloGAN [36] utilizes 3D-CNN to generate low-resolution voxel features and projects them as 2D feature maps for further neural rendering. GRAF [55] adopts a generative radiance field as scene representation and generates images via volumetric rendering [35]. GRAM [12] further sparsifies the radiance field as a set of radiance manifolds and leverages manifold rendering [78, 12] for more realistic image generation. Although 3D-aware GANs are able to control camera viewpoints, they cannot achieve fine-grained control over the generated instances such as shapes and expressions for human faces. This prevents them from being used for multiview face animation generation tasks. In this work, we introduce imitative learning of 3DMM into a 3D-aware GAN framework to achieve expression-controllable face image generation.

**Face image synthesis with 3D morphable model guidance.** 3D Morphable Models (3DMMs) [5, 47, 7] play an important role for face image and animation synthesis [66, 28, 75, 54]. Earlier works [65, 66] directly render textured meshes represented by 3DMM using traditional rendering pipeline for face reenactment. However, these methods often produce over-smooth textures and cannot generate non-face regions due to the limitation of 3DMM. Later works [28, 18, 15, 75, 8] apply refinement networks on top of the rendered 3DMM images to generate more realistic texture details and fill the missing regions. Nevertheless, these methods still rely on a graphics rendering procedure to synthesize coarse images at inference time, which complicates the whole system. Recently, DiscoFaceGAN [11] proposes an imitative-contrastive learning scheme that enforces the generative network to mimic the rendering process of 3DMM. After training, it can directly generate realistic face images and control multiple face attributes in a disentangled manner via the learned 2D GAN. Some concurrent and following works [64, 30, 20, 51] of it share a similar spirit. However, these methods often encounter 3D-inconsistency issues when controlling camera poses due to the non-physical image rendering process of 2D GANs. Some recent methods [16, 24] incorporate 3DMM knowledge into NeRF's training process to achieve animatable face image synthesis. However, they do not leverage a generative training paradigm [21] and require video sequences or multiview images as supervision. By contrast, our method can be trained with only unstructured 2D images.

**Face editing and animation with GANs.** A large amount of works adopt GANs for face image manipulation and animation synthesis [50, 2, 19, 57, 52, 69, 58, 56, 63, 1, 67, 70, 46, 17, 54, 71, 77].

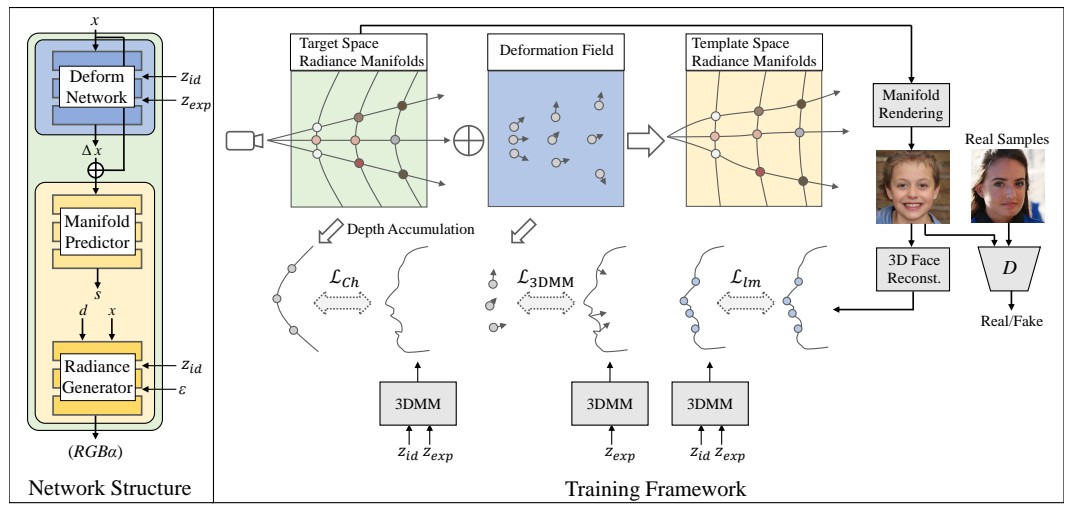

Figure 2: Our proposed framework which consists of a template radiance field and an expression-driven deformation field for animatable 3D-aware face image generation.

However, these methods do not offer a generative modeling of face identities thus are difficult to generate new virtual subjects. They also suffer from 3D-inconsistency and texture flickering issues when changing camera poses due to the use of 2D CNN as image renderer.

## 3   Approach

Our AniFaceGAN is a generative model trained on a collection of unstructured monocular face images. The generator produces two 3D fields, namely a template radiance field and an expression-driven 3D deformation field. Figure 2 shows the overall pipeline of our method. The inputs to our generator include an identity code $\boldsymbol{z}_{id} \in \mathbb{R}^{d_i}$ for shapes, an expression code $\boldsymbol{z}_{exp} \in \mathbb{R}^{d_e}$ for facial animations, an additional noise $\boldsymbol{\varepsilon} \in \mathbb{R}^{d_\varepsilon}$ controlling other attributes such as appearance, lighting and background, and a camera pose $\boldsymbol{\theta} \in \mathbb{R}^3$. We incorporate the 3D prior of a 3DMM face model [47] into our generative modeling. With 3DMM, the geometry of a 3D face represented by a mesh $\mathrm{S} \in \mathbb{R}^{3N}$ of $N$ vertices is parameterized as

$$\mathrm{S} = \mathrm{S}(\beta, \gamma) = \bar{\mathrm{S}} + \mathrm{B}_{id}\beta + \mathrm{B}_{exp}\gamma, \tag{1}$$

where $\mathrm{B}_{id}$ and $\mathrm{B}_{exp}$ are the PCA basis for identity and expression, $\beta, \gamma$ are the identity and expression coefficients, and $\bar{\mathrm{S}}$ is the averaged face shape. Our latent spaces of identity and expression are aligned with the 3DMM model to facilitate our 3D-space imitation learning (to be described later) and expression control. During GAN training, we sample $\boldsymbol{z}_{id}$ and $\boldsymbol{z}_{exp}$ from prior distributions estimated from real images, which is similar to [11].

### 3.1   Template radiance field

The template radiance field determines the geometry and appearance of the generated identity, and is defined as a canonical space with a neutral expression. It is modeled by a template radiance field generator $G$, which takes a 3D point $\boldsymbol{x}$, identity code $\boldsymbol{z}_{id}$, noise code $\boldsymbol{\varepsilon}$, and view direction $\boldsymbol{d}$ as input and output the color $\boldsymbol{c}$ and occupancy $\alpha$ of $\boldsymbol{x}$:

$$G : (\boldsymbol{x}, \boldsymbol{z_{id}}, \boldsymbol{\varepsilon}, \boldsymbol{d}) \in \mathbb{R}^{d_i + d_\varepsilon + 6} \rightarrow (\boldsymbol{c}, \alpha) \in \mathbb{R}^4. \tag{2}$$

In this work, we use the generative radiance manifolds (GRAM) [12] approach to generate our template radiance field. GRAM is a state-of-the-art 3D-aware GAN that can generate high-quality images with strong multiview consistency by regularizing radiance learning and sampling on a set of learned surface manifolds. In theory, our template field can be generated by any 3D-aware generative models. We modify the generator of GRAM by changing its input to ours and train it from scratch within our whole framework.

## 3.2 Expression-driven 3D deformation field

The 3D deformation field is used to generate different expressions by deforming the 3D neural face in the template radiance field. It is represented by a deformation network $F$.

In practice, $F$ models an "inverse" deformation, which deforms a 3D point in the target space to the template space. The input to $F$ includes a target-space 3D point $x$ and expression code $z_{exp}$, and the output is a displacement vector representing the 3D offset from the target space to template space. Since identity and expression are correlated in 3DMM (see their linear combination in Eq. 1), we also add $z_{id}$ to the input of $F$. In summary, $F$ can be written as:

$$F : (\boldsymbol{x}, \boldsymbol{z}_{id}, \boldsymbol{z}_{exp}) \in \mathbb{R}^{d_i+d_e+3} \to \Delta \boldsymbol{x} \in \mathbb{R}^3. \tag{3}$$

## 3.3 Image rendering

Our rendering follows a volume rendering paradigm tailored to the radiance manifold rendering of [12]. For each viewing ray $r$ in the target space, we uniformly sample $N$ points $\{x_i\}$ along the ray and deform them to the template space by $F$ as $\{x_i'\}$ where $x_i' = x_i + \Delta x_i$. Then we calculate the $M$ intersection points between the (deformed) ray and template radiance manifolds of GRAM, denoted as $\{x_j^*\}$. A differentiable intersection calculation method is used as in [12]. Finally, we obtain the color and occupancy of $\{x_j^*\}$ with $G$, and composite the output color via the following rendering equation [40, 78, 12]:

$$C(\boldsymbol{r}) = \sum_{j=1}^{M} T(\boldsymbol{x}_j^*)\alpha(\boldsymbol{x}_j^*)\boldsymbol{c}(\boldsymbol{x}_j^*) = \sum_{j=1}^{M} \prod_{k<j}(1 - \alpha(\boldsymbol{x}_k^*))\alpha(\boldsymbol{x}_j^*)\boldsymbol{c}(\boldsymbol{x}_j^*). \tag{4}$$

Note that the above process is equivalent to first deforming the radiance manifolds in the template space to the target space by a "forward" deformation, and then conducting intersection calculation and manifold rendering in the target space. Here we use the inverse deformation defined by $F$ (Eq. 3) in that the radiance manifolds are defined on a set of implicit surfaces.

## 3.4 3D-space imitation learning

To ensure that the face shape and expression of a generated subject follow those described by 3DMM given input identity and expression codes, we employ a set of 3D-space losses to enforce the geometry and expression deformation obtained by our generator to imitate the 3DMM model.

**Dense geometry imitation.** First, we enforce the similarity between the underlying 3D geometry of our generated instance and its corresponding 3DMM face. To achieve this, we first extract a depth map of the generated instance in the target space. Specifically, for each viewing ray $r$ with sample points $\{x_i\}$, its accumulated depth value can be computed via

$$z(\boldsymbol{r}) = \sum_{i=1}^{N} T(\boldsymbol{x}_i)z(\boldsymbol{x}_i), \tag{5}$$

where $T(\cdot)$ is defined in Eq. 4. We use intersections between $r$ and the manifolds in the target space as the sample points to calculate the depth, as shown in Fig. 2. Then we reproject each pixel to 3D space according to its depth, generating a point cloud S'. We compare S' with a 3D face $S(\boldsymbol{z}_{id}, \boldsymbol{z}_{exp})$ obtained by 3DMM via Eq. 1. Since 3DMM only represents a small facial region, we employ a directed Chamfer distance [61] to align the 3DMM shape to S':

$$\mathcal{L}_{Ch}(\mathrm{S}, \mathrm{S}') = \frac{1}{|\mathrm{S}|} \sum_{\mathrm{x} \in \mathrm{S}} \min_{\mathrm{y} \in \mathrm{S}'} \|\mathrm{x} - \mathrm{y}\|_2^2. \tag{6}$$

**3D landmark imitation.** We then incorporate a 3D landmark loss function to enforce the deformation field to generate desired expressions. Let $I$ be an image generated by our model with identity code $z_{id}$ and expression code $z_{exp}$. We use a face reconstruction network [13] to extract the 3DMM identity and expression coefficients on $I$, denoted as $\hat{z}_{id}$ and $\hat{z}_{exp}$. A 3DMM shape can be reconstructed using $\hat{z}_{id}$ and $\hat{z}_{exp}$. The 3D landmark loss is then defined as

$$\mathcal{L}_{lm} = \|f_{lm}(\mathrm{S}(\boldsymbol{z}_{id}, \boldsymbol{z}_{exp})), f_{lm}(\mathrm{S}(\hat{\boldsymbol{z}}_{id}, \hat{\boldsymbol{z}}_{exp}))\|_2^2, \tag{7}$$

where $f_{lm}$ represents a function of extracting the landmarks. Here we use the landmark points as in [13] except for those on face contour. Note that we define landmark loss in 3D space instead of projecting them on 2D image plane for loss calculation as done in [11] because 2D landmarks have larger ambiguity between different expressions.

**Deformation imitation.** We also encourage the deformation field to follow 3DMM deformation. Specifically, for each 3D point $x$ in the target space, we first find its nearest point $x_{ref}$ on the corresponding 3DMM mesh and compute its deformation to the neutral face according to the 3DMM model defined in Eq. 1, i.e.,

$$\Delta x_{ref} = -\big(S(z_{id}, z_{exp})(x_{ref}) - S(z_{id}, \mathbf{0})(x_{ref})\big) = -B_{exp} z_{exp}(x_{ref}), \tag{8}$$

and then employ a loss function $\mathcal{L}_{3\text{DMM}}$:

$$\mathcal{L}_{3\text{DMM}} = \|\Delta x - \Delta x_{ref}\|_2^2, \tag{9}$$

where $\Delta x$ is the deformation of $x$ obtained via Eq. 3.

**Deformation regularizations.** We expect the deformation of most points in the target space to be small except for regions influenced by expression change. To this end, we employ a minimal deformation constraint via

$$\mathcal{L}_{reg} = \|\Delta x\|_2^2. \tag{10}$$

Moreover, to encourage the deformation fields to be smooth and avoid abrupt deformation changes, we impose a simple smoothness constrain on the deformation field:

$$\mathcal{L}_{smooth} = \|\Delta x - \Delta(x + \xi)\|_2^2, \tag{11}$$

where $\xi$ is a small random perturbation.

**Adversarial learning.** Following [12], we randomly sample real images from the training set, and apply a discriminator $D$ to differentiate synthesized images from real images via the same adversarial loss used in [12].

## 4 Experiments

We train our method on the FFHQ [26] dataset[3] which contains 70K face images. Following [11], we randomly sample latent codes $z_{id}$, $z_{exp}$, and camera pose $\theta$ from estimated prior distributions, and sample $\varepsilon$ from a normal distribution. In our experiments, the Adam optimizer [29] with $\beta_1 = 0$ and $\beta_2 = 0.9$ is applied for training our model. We set the learning rate to $2e-5$ for the deformation network and the generative radiance manifolds, and $2e-4$ for the discriminator. We train our models on 8 Nvidia Tesla V100 GPUs with a batch size of 32 at the resolution of $128 \times 128$. Our model takes approximately three days for training. See the supplement for more details.

### 4.1 Evaluation on generation quality

To evaluate the image generation quality of our method, we compare with two previous face generative models, CONFIG [30] and DiscoFaceGAN [11], which also achieve disentangled control over camera views and expressions of their generated virtual subjects. We further compare with a baseline method we call DiscoFaceGRAM, which naively combines GRAM [12] with DiscoFaceGAN's imitative-contrastive learning scheme. Specifically, in [11], the GAN model is a black box to its imitative-contrastive learning scheme and thus can be replaced by any other GAN model. Thus in DiscoFaceGRAM, we simply substituted the original StyleGAN in [11] with GRAM used in AniFaceGAN while keeping other parts unchanged, and trained this so-called DiscoFaceGRAM using the framework of [11]. For DiscoFaceGRAM, we discard the deformation field and directly send $z_{id}$, $z_{exp}$ to the template field represented by GRAM. Since we do not control textures and illuminations as in [11], we only apply the imitative loss for 2D landmarks and the contrastive loss for expressions to train this baseline method. The purpose of developing such a baseline is to validate the contribution of the 3D deformation and 3D-level imitation scheme we newly designed in this work. For CONFIG and DiscoFaceGAN, we use the pretrained models released by their authors.

---

[3]FFHQ is released under the Creative Commons BY-NC-SA 4.0 license; the human face images therein were published on Flickr by their authors under licenses that all allow free use for non-commercial purposes.

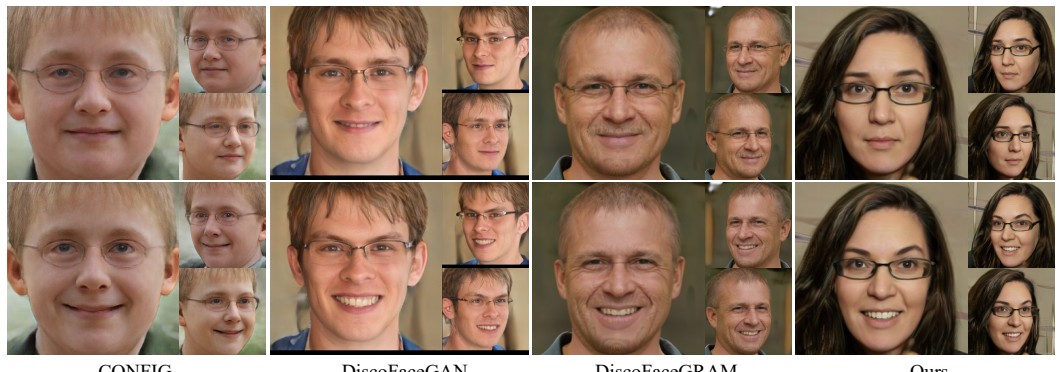

|  CONFIG | DiscoFaceGAN | DiscoFaceGRAM | Ours |

Figure 3: Visual comparison between our method and others. The quality of the images synthesized by CONFIG and DiscoFaceGRAM is clearly lower than ours. Both DiscoFaceGAN and our method can generate photorealistic images. However, DiscoFaceGAN suffers from inconsistency for different expressions and poses. (**See the accompanying video for better visualization and comparison**.)

Table 1: Quantitative evaluation of different models on the FFHQ dataset. FID and KID ($\times 100$) are computed with 5K synthesized images and 5K real images. DiscoFaceGRAM is a baseline method implemented by us (see text for details).

|  | GRAM [12] | CONFIG [30] | DiscoFaceGAN [11] | DiscoFaceGRAM | Ours |
|---|---|---|---|---|---|
| FID↓ | 19.4 | 52.6 | 17.9 | 23.9 | 19.9 |
| KID↓ | 0.64 | 3.38 | 0.79 | 1.19 | 0.86 |

We first show the visual comparison in Fig. 3. CONFIG and DiscoFaceGAN suffer from 3D-inconsistency issues (e.g., see bangs of the hairs) when varying the camera pose as these methods are based on black-box CNN renderers. In DiscoFaceGRAM, the generation process handles expression change without using explicit deformation, and all the losses are imposed on the generated 2D images. We conjecture that such strong 2D-level losses may introduce some hurdles for 3D-aware GAN training, which might have led to its lower image quality. In addition, modeling expressions without deformation also leads to geometry and texture inconsistency (e.g., see the variation of hair bangs and eyeglasses under expression change). By contrast, our method shows strong consistency between generated images of different poses and expressions, thanks to the manifold rendering of our inherent 3D representation and the expression-driven 3D deformation. A more detailed visual comparison in terms of consistency is shown in Fig. 4 (see Sec. 4.3). DiscoFaceGRAM cannot guarantee a full disentanglement between identity and expression (e.g., see the hair, color of face and eyeglasses while changing expressions) using only image-space imitative and contrastive constraints. Instead, our method physically disentangles the two factors and demonstrates better generation quality and more reasonable expression control.

For quantitative evaluation, we compute the Fréchet inception distance (FID) [23] and Kernel Inception Distances (KID) [4] between 5K randomly synthesized images and 5K randomly sampled real images [12]. As shown in Table 1, our result is slightly worse than GRAM, which is expected since vanilla adversarial training focuses only on image quality while we introduce controllability with additional training losses. The quality of our method is also moderately lower than DiscoFaceGAN, which is also reasonable since our backbone GRAM, as a 3D-aware GAN, still has a quality gap to traditional 2D GANs based on the powerful StyleGAN architecture. Compared with GRAM, our method introduces strong controllability over expression with only a slight decrease of image quality. DiscoFaceGAN shows better image quality in terms of FID and KID, but sacrifices 3D consistency which is critical for realistic video generation. The large FID&KID gaps between DiscofaceGRAM and our method further validate the effectiveness of our model and loss design.

## 4.2 Evaluation on factor disentanglement

To further evaluate the disentangled controllability of AniFaceGAN over expression and pose, we calculate the disentanglement score (DS) [11] of generated images. DS measures that, when we

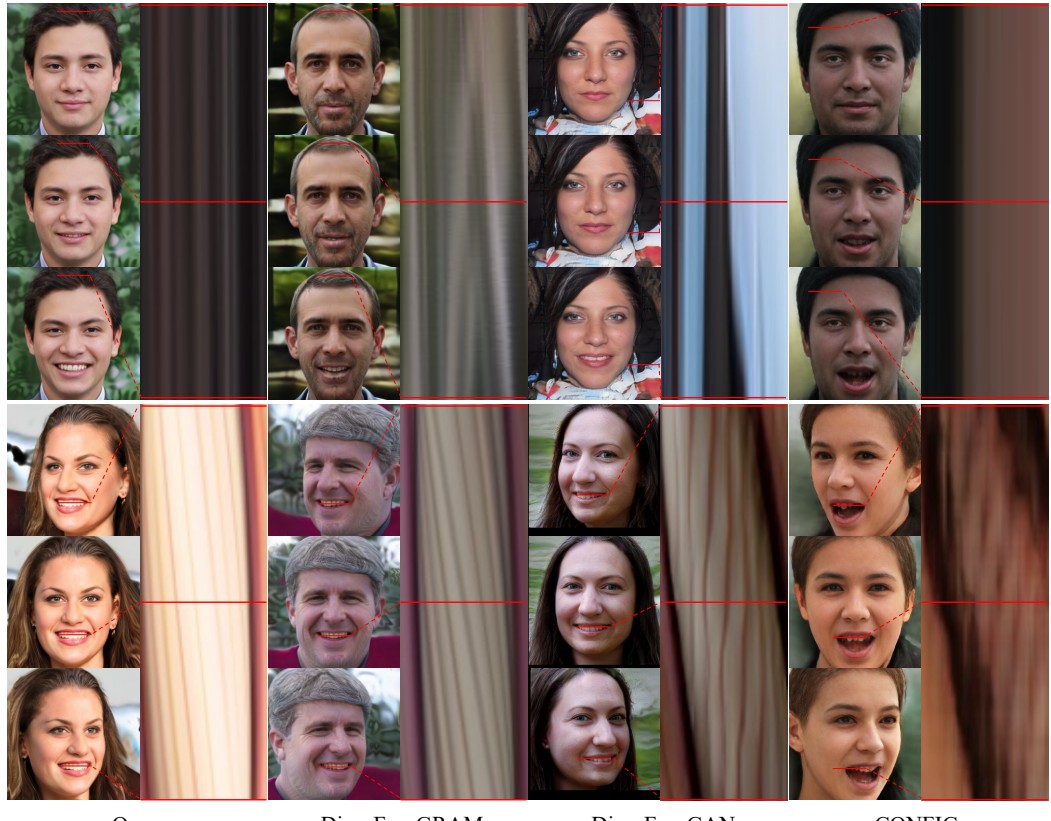

|  Ours | DiscoFaceGRAM | DiscoFaceGAN | CONFIG |

Figure 4: Comparison of consistency using a visualization method similar to [74]. For the first row, we fix the camera location and smoothly change the expression. For the second row, we use the same expression and rotate the camera smoothly. We check the texture of a line segment at a fixed 2D position. Twisted lines in the first group indicate unwanted texture flickering, and the distortion/noise pattern in the second group indicates 3D inconsistency of the rendered multiview images.

Table 2: Evaluation of factor disentanglement on expression and pose using disentanglement score (DS) [11] and 3D consistency using the multiview reconstruction method of NeuS [68]. The disentanglement scores of GRAM [12]+Face-vid2vid [71] is not included as [71], unlike other methods, does not use 3DMM coefficients for expression control.

| Method | Disentanglement | | 3D Consistency | |
| | $DS_{exp} \uparrow$ | $DS_{pose} \uparrow$ | PSNR$\uparrow$ | SSIM$\uparrow$ |
| DiscoFaceGAN [11] | 23.84 | 4.43 | 33.3 | 0.925 |
| GRAM [12]+PIRenderer [54] | 13.50 | 6.06 | 38.1 | 0.971 |
| GRAM [12]+Face-vid2vid [71] | - | - | 38.3 | 0.970 |
| Ours | **24.43** | **7.29** | **41.1** | **0.984** |

only vary one single factor, if other factors of the generated images are stable. We calculate DS for expression and pose following the same experiment setting as [11] and compare the result with [11]. We also compare with another method we call GRAM+PIRenderer, which leverages a state-of-the-art face editing method PIRenderer [54] to modify the generated frontal images of GRAM [12] for expression and pose control. As shown in Table 2, PIRenderer [54] cannot well disentangle identity and expression as indicated by the low $DS_{exp}$, even though it is trained on video data. DiscoFaceGAN achieves better expression disentanglement thanks to its contrastive learning scheme, yet it still suffers from detail inconsistency as shown in Fig. 3 and 4. Our method utilizes 3D deformation for animation and achieves the best disentanglement of expression. It also obtains the highest DS for pose, which validates the advantage of using 3D representation for 3D-consistent image generation.

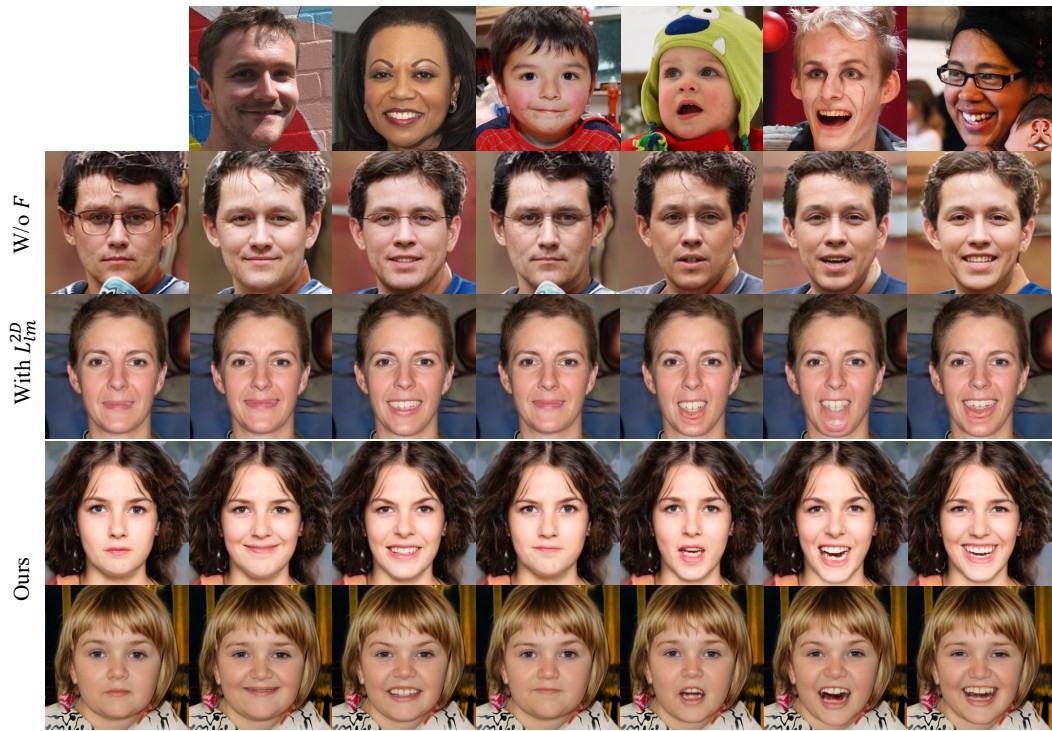

Figure 5: Visual comparison on expression control. The first row shows the real images that provide reference expressions, and other rows are the results from different methods (see text for details).

Table 3: Ablation study on deformation.

| Method | w/o $F$ | Ours |
|---|---|---|
| FID↓ | 28.3 | 19.9 |

Table 4: Ablation study on 3D-space imitation losses.

| Method | w/o $\mathcal{L}_{lm}$ | w/o $\mathcal{L}_{Ch}$ | w/o $\mathcal{L}_{3DMM}$ | with $\mathcal{L}_{lm}^{2d}$ | Ours |
|---|---|---|---|---|---|
| FID↓ | 22.4 | 21.8 | 19.8 | 22.1 | 19.9 |

## 4.3 Evaluation on consistency

Since DS can only evaluate the factor disentanglement on a semantic level, we conduct further experiments to validate the consistency of our method under expression and pose change on a more detailed level. Following the idea of [74], we first present the spatiotemporal textures of different methods by smoothly varying expression or camera pose and stacking the texture of a fixed horizontal line segment (Fig. 4). For disentangled expression control, regions not affected by the expression should remain unchanged, leading to a texture image with vertical strips. For 3D-consistent pose variation, the resultant texture image should be similar to the Epipolar Line Images (EPI) [6], which has smoothly tilted strips. As shown in Fig. 4, our AniFaceGAN produces desirable texture patterns under expression and pose changes, indicating its strong consistency. DiscoFaceGRAM and DiscoFaceGAN produce non-vertical strips when changing expression, indicating inconsistency of detailed appearance. Under pose change, DiscoFaceGAN and CONFIG lead to distorted texture images which indicate the 3D-inconsistency issue.

We further evaluate the 3D consistency of our method quantitatively and compare with state-of-the-art 2D face editing methods PIRenderer [54] and Face-vid2vid [71]. Following [74], we randomly generate 50 identities with GRAM, and then generate 30 multiview images under a fixed set of viewpoints for each identity via different methods. We then train the state-of-the-art multiview reconstruction method NeuS [68] on them and compare the average PSNR and SSIM scores of the images reconstructed by NeuS. In theory, the more consistent the input multiview images are, the higher the reconstruction quality will be. As shown in Table 2, our method has the best performance in terms of PSNR and SSIM, indicating its superiority over other alternatives at preserving 3D consistency. Also note that our method is a generative modeling method which can synthesize virtual identities, while PIRenderer and Face-vid2vid cannot generate non-existing subjects solely. The visual results can be found in the supplement.

### 4.4 Ablation study

We conduct ablation studies to evaluate the efficacy of each component of our framework. The results are shown in Table 3, 4 and Fig. 5. We train each variant five times using different random seeds and present the average score. In Table 3, we compare our method with a baseline method without using the deformation field but directly sending $z_{id}$ and $z_{exp}$ into the template GRAM (*w/o F*). As shown, the image generation quality without the deformation field degrades by a large margin. The corresponding visual results in Fig. 5 also depict that removing the deformation leads to a loss of disentanglement between identity and expression.

In Table 4, we validate the efficacy of our proposed 3D-level imitations by removing each loss component during training (w/o $\mathcal{L}_*$). We also conduct an experiment which replaces the 3D landmark loss with its 2D counterpart (with $\mathcal{L}_{lm}^{2d}$) by calculating landmark loss on the projected image plane as in [11]. As shown, removing $\mathcal{L}_{3DMM}$ yields a similar FID score, but we empirically find our full model has slightly better visual quality. Removing other loss components all lead to degradation of generated image quality. In addition, replacing our 3D space landmark loss with 2D landmark loss also degrades the image quality and introduces obvious artifacts for certain expressions as shown in Fig. 5. We conjecture this is due to the misalignment between the 2D position distributions of the projected 3DMMs and the faces in real images, which causes conflicts between the 2D landmark imitative loss and the adversarial loss. In contrast, our full model yields high-quality image generation results with accurate control over expression.

### 4.5 Animatable face video generation

Although our method is trained on unstructured 2D images, it can be used to generate realistic video animations of virtual subjects. We achieve this by generating image sequences with our pre-trained AniFaceGAN using continuous expression codes and camera poses (either handcrafted ones or those extracted from real videos). Thanks to our proposed 3D representation of combined template field and expression-driven deformation field as well as the carefully designed imitative losses, we are able to generate high-quality face videos with strong geometry and appearance consistency under continuous expression and pose changes, which cannot be achieved by previous methods. More details and visual results can be found in the supplement and accompanying video.

## 5  Conclusions

We have presented an animatable 3D-aware face image generation method named AniFaceGAN, which aims to synthesize realistic face images with controllable poses and expressions. To maintain 3D consistency across different expressions, we leverage 3D deformation to derive the desired expressions. For face image rendering, we propose to generate two 3D fields, i.e., a template radiance field and an expression-driven deformation field. A set of novel 3D-space imitation losses are proposed to effectively train our model along with adversarial learning. The proposed AniFaceGAN is an unconditional generation model that is trained on unstructured 2D face images without any dependency on 3D or multiview data. Experimentally, our method trained on FFHQ dataset can produce high-quality animatable faces with strong visual 3D consistency across different poses and expressions. We hope our method could serve as a strong baseline for realistic face video generation and animation.

## 6  Broader Impact

This work aims to design an animatable 3D-aware face image generation method for the application of photorealistic virtual avatars. It is not intended to create content that is used to mislead or deceive. However, like other related face image generation techniques, it could still potentially be misused for impersonating humans. We condemn any behavior to create misleading or harmful content of a real person. Currently, the images generated by this method contain visual artifacts that can be easily identified. The method is data-driven, and the performance is affected by the biases in the data. So one should be careful about the data collection process when using it.

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
