# OpenReview forum: "AniFaceGAN: Animatable 3D-Aware Face Image Generation for Video Avatars"
_NeurIPS.cc/2022/Conference — NeurIPS 2022 Accept_

### Official Review · Reviewer_ch4r · 2022-06-23

**Rating:** 4
**Confidence:** 4
**Soundness:** 3 good
**Presentation:** 2 fair
**Contribution:** 2 fair

**Summary:**

This paper proposed a framework for face image manipulation. It attempts to achieve fine-grained control over attributes and facial expressions by better preserving 3D consistency. The key contributions include a template implicit field and a 3D deformation field. The reported experimental results show that the proposed method can produce high-quality animatable video avatars with good texture consistency.

**Questions:**

1. What is the motivation behind AniFaceGAN? In introduction, there is not enough discussion about the motivation. From line 37 to line 38, the transition is not smooth. What weakness does NeRF have and how the proposed AniFaceGAN can do better? If NeRF is already "producing realistic images with strong 3D consistency", why do authors bother inventing a new method?
2. What does Fig. 2 try to express? I had thought this is the overall framework but apparently, it is at most incomplete. For face image manipulation, we expect both input and output are face images and the underlying 3D model serves as prior or regularization terms. The training framework on the right and the network structure on the left do not help each other. It is simply too difficult to decipher how this figure is related to the task of face image manipulation - e.g., what is the difference between the two radiance manifolds? how come the arrow points from target to template (not the other way around)?
3. Regarding Figs. 3-4, why do authors use different face identities in visual comparison? The same question applies to the video in supplementary materials. When different faces are used to compare different methods, it is difficult to see the difference.
4. About the comparison with DiscoFaceGAN, there are a few strong claims - e.g., "DiscoFaceGAN shows better image quality in terms of FID and KID, but sacrifices the 3D-consistency" (line 218) and "DiscoFaceGAN suffers from inconsistency for different expressions and poses" (caption of Fig. 5). I am afraid I cannot see this weakness from either Fig. 5 or supplementary video. Can authors provide further clarification? Note that something obvious to the authors might not be that apparent to others.
5. How about the computational complexity of AniFaceGAN? There is little discussion about the computational cost.

**Ethics Review Area:**

["I don’t know"]

**Limitations:**

I am curious about the performance of AniFaceGAN in some extreme situations - e.g., extreme pose or lighting conditions. Meanwhile, all demo video are very short - it is difficult to evaluate the quality of an animation for such short duration. I am wondering if the authors have video demo of longer period - say >3 seconds.

**Strengths And Weaknesses:**

Strengths:
+ The concept of 3D-aware face generation is a sensible idea. Ensuring 3D consistency for face image manipulation has shown high texture consistency across poses and expressions.
Weaknesses:
- It applied 3DMM expression parameters as a prior for disentanglement of latent variables. Although the experiments in this paper only focuses on 2D face image editing, from the title to the abstract, it easily confuses the reader that they addressed 3D face editing task - e.g., animation usually implies the generation of a 3D avatar from a given face.  The last sentence of the abstract about "strong visual 3D consistency" is misleading because there is no 3D reconstruction and editing contribution in this paper indeed.
- Regarding the 2D face image generation task, the idea of combining image rendering and 3DMM guided face editing is not novel. Also, from the experimental result view, they are not impressive to argue their contributions. From Figure 3 to Figure 4, their improvements are marginal compared to DiscofaceGAN.
- More importantly, the comparisons are not convincing with only two out of dated methods.  Some missing methods will need to be considered for comparison in the experiment:
TransEditor: Transformer-Based Dual-Space GAN for Highly Controllable Facial Editing
InterFaceGAN: Interpreting the Disentangled Face Representation Learned by GANs

---

> ### Author Response · Authors · 2022-08-02
> **Responsed to Reviewer ch4r (Part 1)**
>
> We thank the reviewer for the comments and suggestions.
>
> To help the reviewer better understand our task and method in this paper, we think it might be necessacy to introduce some concepts here and rephrase our introduction. First, NeRF is a scene representation developed for synthesing novel views of a scene; see [35]. Given multiview images of a single scene, NeRF can be trained to model the radiance field of this scene and render novel view images after training. NeRF is NOT used to generating novel scenes/objects. GANs, in our context, are used to generate images of novel, non-existing objects; see [21]. The training data for GAN is a collection of unstructured 2D images, each from one object. Recently, 3D-aware GANs emerged, which use the NeRF as their underlying representation for generative modeling of real images; see [55,10,22,12,38,9]. "3D-aware" is a commonly used term in this field, meaning that the generator is aware of the underlying 3D geometry, and can control the 3D camera viewpoints of generated images; see [55,10,22,12]. Even though there's no explicit "3D reconstruction", 3D consistency is the key advantage of 3D-aware GANs, meaning the images of a generated novel object are visually consistent when rendered at different views; again, please see [55,10,22,12].
>
> The task this paper deals with is to introduce controllability into 3D-aware GAN trained on human faces. As stated in the abstract and introduction, existing 3D-aware GANs do not offer control over important facial attributes such as expression. So our motivation is to introduce face expression control into a NeRF-based 3D-aware GAN. Our method is a GAN, which means it is a generative modeling method trained to synthesize novel objects. It is NOT a 2D face image manipulation method. Almost all our results shown in the paper and supplementary video, expect the very last one in the video, are the results of virtual faces directly generated by our trained AniFaceGAN. They are not created by manipulating existing images.
>
> We now respond to the reviewer's questions.
>
> - **Motivation.** As mentioned above (and also in the abstract and introduction), existing
> 3D-aware GANs using NeRF representation have shown promising image generation results and their 3D consistency is very appealing for generating virtual video avatars. However, they do not have control over face expressions. Our motivation is to introduce facial expression control into 3D-aware GAN so that we can animate the generated virtual faces. Our contributions include a 3DMM-guided radiance field deformation scheme and the 3D-level imitation losses.
>
>     - "*What weakness does NeRF have and how the proposed AniFaceGAN can do better?*" "*If NeRF is already "producing realistic images with strong 3D consistency", why do authors bother inventing a new method?*" As stated, NeRF is a scene representation and novel view synthesis method for a single scene/object. Our method is a GAN using NeRF as the underlying representation. It learns to generate images of novel, non-existing objects from single images of many real objects. These two methods handle different tasks and they are simply not comparable.
>
> - **Fig. 2 and the overall framework.** As stated, our method is NOT a face image manipulation one. Fig. 2 depicts our network structures and training framework. As Fig. 2 (right) shows, the input to our method is a random noise \epsilon, an identity code z_id, an expression code z_exp, and a rendering viewpoint \theta. These vectors are randomly sampled during training. Then our generator generates a set of (deformed) radiance manifolds which are used to render an face image. The discriminator take both the generated and real images as input and judge whether they are real or fake. The generator and discriminator play a min-max game as in standard adversarial learning scheme; see [21]. As illustrated in Fig. 2, we also apply several 3D-level imitation losses to incorporate the 3DMM prior and gain controllability over expression. After GAN training, we can use the generator to generate a new virtual identity by randomly sampling z_id and \epsilon, and get desired expression and pose by controlling z_exp and \theta.
>
> - **Why using different identities for Fig 3,4 and the supplementery.** Again, our method and the compared ones in the paper (CONFIG, DiscoFaceGAN, DiscoFaceGRAM) are GAN models. The identities they generate are random and one cannot get same identiies using them. We understand that comparing the results with different identities is difficult, but till now there's no better visual comparison strategy and this is perhaps still the most commonly used one.

---

> ### Author Response · Authors · 2022-08-02
> **Responsed to Reviewer ch4r (Part 2)**
>
> - **Comparison with DiscoFaceGAN.** The 3D inconsistency of DiscoFaceGAN should be easily observable from our supplementary video. We invite the reviewer to re-check 01'35''- 02'06''. It can be observed that when the persons talk and change pose, the textures on the rendered images are very unnatural and unstable. The most prominent region might be hair, where one can clearly observe the hair style changes in DiscoFaceGAN's results. Upon closer and more careful visual inspection, one may also find that the texture of other facial components also changes across time. A good result with high 3D consistency should make the observer feel that he/she is just viewing an object in 3D, such as the results of our AniFaceGAN shown on the right.
>
>     The superior performance of our method compared to DsicoFaceGAN as well as other baselines is not only due to the 3D-aware GAN we used, be also because of the 3D deformation scheme and 3D-level imitation losses we newly designed in paper.
>
> - **Computational cost.** Currently, our unoptimized implementaion of AniFaceGAN takes 0.84 seconds to render a image of resolution 256x256. We will add a discussion about the computational cost in the revised paper.
>
> - **Limitations and longer video clips.** AniFaceGAN can generate images of decent quality with yaw angles within 45 degrees. For yaw angles larger than 60 degrees, it produces stratified artifacts due to the underlying radiance manifolds representation. Note that AniFaceGAN is not a face editing method, and it does not explicitly model lighting. As it is a GAN method, it also has difficulty in generating images with lighting conditions out of the training data distribution.
>
>     We are confused by "*all demo videos are very short*", "*if the authors have video demo of longer, say >3 seconds*". Our supplementary video contains multiple animation clips of over 30 seconds, and there's no very short clips of less than 3 seconds. We are not sure if the reviewer has been looking into the right video file for our paper, and we kindly ask the reviewer to double check it. We also invite the reviewer to check more of our video demos at these anonymous links: [video_lipmovement_w_audio_1.mp4](https://user-images.githubusercontent.com/68846118/183392192-61478b5a-364e-4adb-84d6-c2b5aa5d688b.mp4), [video_lipmovement_w_audio_2.mp4](https://user-images.githubusercontent.com/68846118/183392279-9e798f37-7706-409f-9482-9e9553bca5d8.mp4)
>     [video_compare_w_interfacegan.mp4](https://user-images.githubusercontent.com/68846118/182584532-4f85dea9-13f7-480a-9350-7a8eddf15146.mp4). (if the browser does not support online video playing please download and play them locally)

---

### Official Review · Reviewer_FGbb · 2022-07-11

**Rating:** 5
**Confidence:** 4
**Soundness:** 3 good
**Presentation:** 3 good
**Contribution:** 3 good

**Summary:**

This paper proposes a novel pipeline for animatable 3D-aware face image generation. They first propose a template radiance field to obtain the rough face geometry, then an expression-driven deformation field is proposed to animate the facial images based on the change of 3DMM parameters. Qualitative results verify the effectiveness of the proposed pipeline.

**Questions:**

The questions are elaborated in the weakness section, see above.

**Limitations:**

They are all properly discussed in the appendix.

**Strengths And Weaknesses:**

# Strengths

1. The proposed method makes sense to me. It is natural to predict the non-rigid facial movement by inferring the per-point deformation based on 3DMM parameters. The presented qualitative results are mostly visibly plausible.

2. The paper is well-written with thorough related works.

# Weakness

1. In Eq. 7, the chamfer distance is computed between the canonical 3DMM shape (reconstructed by the input exp and id param) and the predicted shape (inferred from the depth map as indicated by authors). How do the authors guarantee that the two shapes are at the same resolution, i.e., they are comparable for CD calculation. Besides, since the 3DMM only contain meaningful information at the facial regions, how do the authors deal with the neighbor regions like hairs?

2. I am curious about how do the authors deal with the wearings, like the glasses, ear rings, etc. Such small items could not be inferred from the 3DMM prior, so how to guarantee the visual quality and 3D shape of them?

3. Missing some relevant comparison baselines. Some important baselines are not compared, including but not limited to [1, 2, 3, 4]

[1] Pie: Portrait image embedding for semantic control, Tewari et al., TOG 2020.

[2] Stylerig: Rigging stylegan for 3d control over portrait images, Tewari et al., CVPR 2020.

[3] Designing an encoder for stylegan image manipulation, Tov et al., SIGGRAPH 2021.

[4] Headnerf: A real-time nerf-based parametric head model, Hong et al., CVPR 2022.

---

> ### Author Response · Authors · 2022-08-02
> **Response to Reviewer FGbb**
>
> We thank the reviewer for the valuable comments and suggestions. Our response can be found below.
>
> - **Chamfer distance computation and handling non-face regions.** As mentioned in Line 165, we employed a *directed* Chamfer distance loss, where we only consider the nearest neighbor point in the reprojected point cloud S' for each point on the 3DMM mesh (no the other way around; see Eq. 7).  There's no resolution issue as the point cloud, generated by lifting every 2D pixel to 3D,  is dense enough. For non-face regions, we simply did not apply any explicit imitation loss. We rely on the adversarial learning and some weak deformation regularizations (minimal deformation constraint in Eq. 10 and deformation smoothness loss in Eq. 11) to handle them.
>
> - **Dealing with wearings.** Again, we did not explicit deal with wearings like glasses, ear rings, etc. Yes, those items could not be inferred from the 3DMM prior. But they actually do NOT need the 3DMM prior, as there should be almost no deformation on them for different expressions. So, as long as our backbone 3D-aware GAN (i.e., GRAM) can learn to generate them using adversarial learning, AniFaceGAN should also be able to deal with them well.
>
> - **Comparison with more methods such as [1][2][3][4].** We thank the reviewer for pointing out these works which we shall cite in our revised paper.
>
>     - First, we would like to reemphasize our method is a GAN with certain desired controllability, and we aim to generate some non-existing faces which could potentially be used as privacy-free video avatars. Our goal of  “*controllable virtual face generation*” is different from "*embedding and editing real faces*", the task studied in PIE[1], StyleRig [2], and Tov et al. [3]. Although we also show the possibility of real image editing with a few examples, it is clearly not our main focus, and high-quality image embedding and editing with 3D-aware GAN is still an open problem. The goal HeadNeRF [4], i.e., building a NeRF-based parametric head model using multi-view multi-expression faces, is also significantly different from ours.
>
>     - Second, it should be clear that these 2D GAN/CNN based editing methods cannot guarantee 3D consistency. The works of [1][2][3] are based on 2D StyleGAN. HeadNeRF [4] uses volumetric rendering for generating intermediate 2D features, but still relies on 2D CNN for feature-to-image translation.
>
>     - Third, in practice, we find it difficult to compare with these four methods: [1][3] do not provide explicit 3D control (e.g., precise 3D pose and targeted expression), and [2] does not provide a publicly available implementation. [4] only handles head region while ignoring others such as neck and background.
>
>     Here we compared to another two face manipulation methods Face-vid2vid (Wang et al. CVPR 2021) and PIRenderer (Ren et al. CVPR 2021) . We invite the reviewer to check our video results at the anonymous links:[video_compare_w_faceediting_1.mp4](https://user-images.githubusercontent.com/68846118/183392379-584672bb-928c-4256-b62b-6af987106da4.mp4),[video_compare_w_faceediting_2.mp4](https://user-images.githubusercontent.com/68846118/183421860-4a9bea68-58f2-4358-a9ef-7dc32d3ba14c.mp4), [video_compare_w_faceediting_3.mp4](https://user-images.githubusercontent.com/68846118/183394174-40d833cf-88e7-45c6-ad4a-417a0626c684.mp4). (If the browser does not support online video playing please download and play them locally) As we can see, these 2D GAN based face manipulation methods cannot guarantee texture consistency. Visually inspected, they cannot well preserve identity and facial geometry under pose change. Another prominent failure of them is eyeglasses, for which our method handles quite well. We also added the comparison with InterFaceGAN, a latent space manipulation method. The video can be found at the anonymous link: [video_compare_w_interfacegan.mp4](https://user-images.githubusercontent.com/68846118/182584532-4f85dea9-13f7-480a-9350-7a8eddf15146.mp4). Again, InterFaceGAN clearly underperforms in terms of 3D consistency. We are happy to include these comparisons in our revised supplementary videos.

---

> > ### Comment · Reviewer_FGbb · 2022-08-08
> > **Response to authors**
> >
> > I thank the authors for the detailed reply. Considering the merit and demerit of this paper, as well as the comments and discussions from other reviewer colleagues, I maintain my score for this paper.

---

> > > ### Author Response · Authors · 2022-08-09
> > > **Thanks for your response**
> > >
> > > We sincerely thank you for the reviews and comments. We have provided corresponding responses and results, which we believe have covered your concerns. We hope to further discuss with you whether or not your concerns have been addressed. Please let us know if you still have any unclear parts of our work. We would be happy to address any follow-up questions.

---

### Official Review · Reviewer_PWhM · 2022-07-13

**Rating:** 5
**Confidence:** 4
**Soundness:** 4 excellent
**Presentation:** 2 fair
**Contribution:** 3 good

**Summary:**

This paper proposes to generate 3DMM controllable 3D-aware faces. They design a generator that produces two 3D fields, namely a template radiance field and an expression-driven 3D deformation field. The expression-driven 3D deformation field is learned with the guidance of 3DMM. Experiments show that this method can produce high-quality 3D consistent controllable faces with delicate control.

**Questions:**

1. The authors write in the "Face editing and animation with GANs" section that ", these methods do not offer a generative modeling of face identities thus are difficult to generate virtual subjects". More discussions could be made. Why not compare with them by first generating virtual samples and then controlling the results with 2D GANs? Or as the authors have shown inversion plus editing results. The comparisons can also be made.

2. Why would the 3D landmark losses affect the FID? As described,  it is computed on 5K randomly synthesized images. And why replacing the 3D landmark loss in Eq. 8 with the 2D landmark regression loss would greatly increase the FID score?

**Limitations:**

1. The whole story in the introduction and title seems to be overclaiming.

Words such as "Realistic Video Avatars" and “high-quality animatable video avatars” have appeared in the title and abstract. From the perspective of Realistic Video Avatars, the results can barely meet the need. As all samples are aligned at the same position, the results are far from realistic.

Moreover, if the authors intend to show results in a reenactment setting, it is better that they can show comparisons with results generated in a [GRAM + Face Vid2vid/PIRenderer] manner, and ask users to provide studies on the realness and quality of the results.

The reviewer's suggestion is to tone down the story and change the title to something like "Realistic Animatable 3D-Aware Face Image Generation with XXX (Explicitly Deformable Radiance Manifolds)"

2. The comparisons and discussions are not sufficient.

a) The authors write that "Perhaps the most relevant work to ours is DiscoFaceGAN". This might be leading readers to believe that only comparing to DiscoFaceGAN is sufficient. Actually, StyleRig is also very relevant to the task of disentangled generative modeling and basically shares the same setting as this paper. The authors should discuss StyleRig (Although they also suffer from texture stitching) and try making a comparison with them.

b) Please also refer to the question part.

c) The numerical comparisons are limited. Why not use the metrics in DiscoFaceGAN on disentanglement for evaluation?

3.  No audio is provided in the supplementary video, thus the reviewer cannot tell whether the lip movements are consistent with the source. Also, there are no 3D results shown, making the 3D consistent claim not strong enough.

**Strengths And Weaknesses:**

++ The demo video of this paper is impressive from the perspective of controllable 3d-aware generative models.

++ The design of one template radiance field and a deformation field is interesting and reasonable.

++ The paper is well-written.

---

> ### Author Response · Authors · 2022-08-02
> **Response to Reviewer PWhM (Part 1)**
>
> We thank the reviewer for the valuable comments and suggestions. Our response can be found below.
>
> - **Comparison with “Generation then Controlling with 2D GAN”.**  First, we’d like to emphasize that this work studies introducing controllability into a (3D-aware face) GAN, which is an important and longstanding research direction. We focus on *generative modelling* certain desired controllability. From the technical perspective, it is fundamentally different from the scheme of “generation with a GAN and controlling it with another GAN”. For this latter scheme, the key lies in the “controlling with another GAN” part, which is sometimes called image editing or face reenactment. Many methods along this direction, such as Face-vid2vid and PIRenderer, *need paired images or videos for training*. Comparing our GAN method trained simply on a collection of unstructured still images with these methods is not entirely fair.
>
> 	Still, per the the reviewer’s suggestion, we tested combining GRAM with face manipulation/reenactment methods, i.e., GRAM + Face-vid2vid and GRAM + PIRenderer. The video results can be found at the anonymous links: [video_compare_w_faceediting_1.mp4](https://user-images.githubusercontent.com/68846118/182415632-796ab0e4-1ee8-4d4c-a711-370789771498.mp4), [video_compare_w_faceediting_2.mp4](https://user-images.githubusercontent.com/68846118/182415740-282e693d-5c3a-4916-b2d0-77c29c5e9e1c.mp4), [video_compare_w_faceediting_3.mp4](https://user-images.githubusercontent.com/68846118/182416048-79e7456d-5576-414a-9f12-6158f1d443d9.mp4). (If the browser does not support online video playing please download and play them locally). As we can see, these 2D CNN based methods cannot guarantee texture consistency. Visually inspected, they cannot well preserve identity and facial geometry under pose change, whereas our deformed 3D radiance manifold representation can generate multiview consistent textures. One prominent failure of them is eyeglasses, for which our method handles quite well. We are happy to include these comparisons in our revised supplementary videos.
>
> - **Impact of landmark losses on FID.** Adding different losses to the generator will very likely affect adversarial learning and hence FID. According to our experiment, adding the 3D landmark loss slightly decreased FID, but the difference is minor. In contrast, the 2D landmark leads to obvious FID increase. As mentioned in Section 4.2, our interpretation is that it is due to unavoidable differences between our defined 3D coordinate system and that of the real faces in the training dataset, there could be certain misalignment of the 2D position distributions between the projected 3DMMs and the real faces, which results in conflicts between the 2D landmark imitative loss and the adversarial loss. Using our 3D landmark loss defined in Eq. 8 does not have this issue because the loss is *totally independent of 3D pose and camera projection* and *avoids any potential misalignment between distributions of our 3DMM projections and real images*.
>
> - **Claim of “video avatars”.** Compared to existing methods and other 2D GAN based solutions (GRAM + Face Vid2vid and GRAM + PIRenderer) shown above, we believe that our method has demonstrated clear advantage in preservation of identity, geometry, and stableness of texture in the generated facial animations, which are highly important for realistic video avatars. Also, our method doesn't need any video for training. Indeed, there are still several limitations preventing the method being immediately applied in real scenarios. We can change the wording in the title to “*towards* Realistic Avatars” to tone down the claim. The title “*AniFaceGAN: Animatable 3D-aware Face Image Generation*” is also good to us.

---

> > ### Comment · Reviewer_PWhM · 2022-08-05
> > **Response to Rebuttal Part 2**
> >
> > Thank the authors for providing the comparisons. However, there are serious issues with the results and I strongly suggest the authors check the code. The experiments on Face Vid2vid are definitely not correctly performed. You can refer to FOMM's code https://github.com/AliaksandrSiarohin/first-order-model to check the problem and comparing your method with FOMM is also acceptable. Please make sure that you are using Face vid2vid but not FS vid2vid. You can use this code for reproduction https://github.com/zhanglonghao1992/One-Shot_Free-View_Neural_Talking_Head_Synthesis.
> >
> > In face reenactment studies, it is inappropriate to center-align all frames. If the driving video has long-span movements, you may try aligning them according to one fixed point instead of performing affine transformations.
> >
> > To the reviewer's understanding, the need for using paired images or videos for training is actually not a limitation. Videos are for now commonly available just as images. It would be good if the authors could try involving video data in this method, which might step further toward realistic avatar.

---

> > > ### Author Response · Authors · 2022-08-08
> > > **Response to Reviewer PWhM (Part 2)**
> > >
> > > We thank the reviewer for the further comments. We updated the results of Face vid2vid using the code repo suggested by the reviewer, and they indeed improved a lot (see updated videos at [video_compare_w_faceediting_1.mp4](https://user-images.githubusercontent.com/68846118/183392379-584672bb-928c-4256-b62b-6af987106da4.mp4),
> > > [video_compare_w_faceediting_2.mp4](https://user-images.githubusercontent.com/68846118/183421860-4a9bea68-58f2-4358-a9ef-7dc32d3ba14c.mp4), [video_compare_w_faceediting_3.mp4](https://user-images.githubusercontent.com/68846118/183394174-40d833cf-88e7-45c6-ad4a-417a0626c684.mp4)). We feed the raw driving video to the code and the image cropping and alignment are handled by itself.
> > >
> > > Despite of the improvements, the results of Face-vid2vid still contain appreciable geometry distortions under pose change, and our method excels at preserving 3D consistency in the face animation videos. To verify this quantitively, we randomly generate 50 identities with GRAM, and then for each identity and each method we generate 30 multiview images under a fixed set of viewpoints. We train the state-of-the-art multiview reconstruction method NeuS [a] on them and compare the average PSNR and SSIM scores of the images reconstructed by NeuS. In theory, the more consistent the input multiview images are, the higher the reconstruction quality will be. As shown by the table below, our method has clearly better performance.
> > >
> > > |Method| PSNR  | SSIM |
> > > |:----:|:-----------:|:-----------:|
> > > |GRAM + PIRenderer |  35.9  | 0.945 |
> > > |GRAM + Face-vid2vid | 38.4 | 0.968 |
> > > |*AniFaceGAN*  |  **40.3**  | **0.978**|
> > >
> > >
> > > We thank the reviewer's suggestion of using videos for our training and will explore it in our future work.
> > >
> > >
> > > [a] P. Wang et al. NeuS: Learning Neural Implicit Surfaces by Volume Rendering for Multi-view Reconstruction. NeurIPS 2021. https://github.com/Totoro97/NeuS

---

> ### Author Response · Authors · 2022-08-02
> **Response to Reviewer PWhM (Part 2)**
>
> - **Comparison with other methods such as StyleRig.**  DiscoFaceGAN is indeed the most relevant work, to the best of our knowledge. StyleRig is relevant, but with due respect, we disagree that StyleRig “basically shares the same setting as this paper”. The reasons are partially discussed in our response to the previous comments. We further summarize the differences below.
>     - *Goal difference.* As the titles of two papers indicate, our main goal is “*generation*”, whereas StyleRig focuses on “*rigging StyleGAN*” (i.e., for *editing*). Specifically, we aim to generate some *random, non-existing* faces which could potentially be used as privacy-free virtual avatars. In contrast, StyleRig aims to provide 3D editing over a given *real photo*. Although we also show the possibility of real image editing with a few examples in the supplementary materials, it is clearly not our main focus, and high-quality image embedding and editing with 3D-aware GAN is still an open problem.
>     - *Method difference.* Due to the different goals mentioned above, the two methods have significantly different setups. Our method, no matter what control we want to incorporate, is still a GAN with an image generator and a discriminator. In contrast, StyleRig is essentially a GAN inversion and latent code editing method requiring a pre-trained GAN. Its key component is a shallow MLP which modifies the inverted latent code according to the given semantic control (See Section 6 of StyleRig).
>     - *Training data difference.* As in a typical GAN setting, we train our generator and discriminator with adversarial learning on a collection of unstructured 2D images. By contrast, StyleRig takes a pretrained GAN model and uses a set of face images and their corresponding latent codes as paired training data (See Section 5 of StyleRig).
>
>     Comparing AniFaceGAN and StyleRig is possible, but it’s difficult to make a fair comparison due to these differences. Moreover, there’s no publicly available implementation of StyleRig, making exact reproduction of their results tricky. As previously mentioned, we made the comparison with GRAM+Face vid2vid and GRAM+PIRenderer, which are also based on 2D CNNs. Here we further compare with GRAM+InterFaceGAN which has a more similar setup with StyleRig, i.e., 2D GAN inversion+editing. The results can be found at the anonymous link: [video_compare_w_interfacegan.mp4](https://user-images.githubusercontent.com/68846118/182584532-4f85dea9-13f7-480a-9350-7a8eddf15146.mp4). Again, GRAM+InterFaceGAN clearly underperforms in terms of 3D consistency.
>
> - **Disentanglement evaluation.** We added the disentanglement evaluation according to the reviewer’s suggestion. We compared the disentanglement score (DS) for expression and pose and followed the respective experiment settings as in DiscoFaceGAN (we did not compare identity as we found the metric to be problematic for it: different methods may generate largely different identity variations, making the numerators in its Eq. 9 not comparable). We compared the results of GRAM+PIRenderer, DiscoFaceGAN, and our AniFaceGAN. As shown in the table below, our method and DiscoFaceGAN have very close disentanglement scores for expression (note that this metric can hardly measure the consistency of texture details), and GRAM+PIRenderer clearly underperformed. Our disentanglement score for pose is much better than DiscoFaceGAN and GRAM+PIRenderer.
>
>     |Method| *DS* (Expression)    | *DS* (Pose) |
>     |:----:|:-----------:|:-----------:|
>     |GRAM + PIRenderer |  11.07  | 5.44 |
>     |DiscoFaceGAN | 23.84 | 4.43 |
>     |*AniFaceGAN*  |  **24.43**  | **7.29**|
>
> - **Results with audio and 3D.** Our driving videos are taken from the 300VW dataset, which unfortunately does not have accompanying audio. For better lip movement quality checking, we generate some virtual talking faces using new source videos with synchronized audio and limited head pose variations. The result videos can be found at the anonymous link: [video_lipmovement_w_audio_1.mp4](https://user-images.githubusercontent.com/68846118/182584627-1e619e76-9cb8-4cd4-9261-f644f8ebae06.mp4), [video_lipmovement_w_audio_2.mp4](https://user-images.githubusercontent.com/68846118/182413052-5afecc27-316c-4b91-8b31-684624008822.mp4), and we are glad to hear the reviewers’ further comments. Besides lip movements, we also would like to invite the reviewer to check the natural, smooth nasolabial folds changes during talking. We are not sure what the reviewer meant by “*no 3D results shown*”. Did the reviewer mean extracting and visualizing the proxy 3D shapes as done in GRAM? If so, our shape quality is identical to GRAM. However, instead of checking 3D shape, we believe the best way to check 3D consistency is to inspect the rendered images under continuous pose change, which is shown in almost all our examples in the video.

---

> > ### Comment · Reviewer_PWhM · 2022-08-05
> > **Response to Rebuttal Part 1**
> >
> > Thank the authors for providing the rebuttal.
> >
> > It is nice for the authors to provide the video results with audio. However, 1) we can barely say that the results provided by Synthesia are "real videos". 2) The results with audio provided clearly demonstrate that this model still cannot capture mouth movement details. To name one "avatar-level" research: https://gafniguy.github.io/4D-Facial-Avatars/. Maybe the authors could discuss this as their limitations.
> >
> > I am not arguing this to be an important problem, but it strengthens my belief that this research is still far from producing Realistic Video Avatars.

---

> > > ### Author Response · Authors · 2022-08-08
> > > **Response to Reviewer PWhM (Part 1)**
> > >
> > > We thank the reviewer for the further comments.
> > >
> > > For 1), we made a mistake in the video and now we have removed the word "real" in those results. Please check the updated videos at [video_lipmovement_w_audio_1.mp4](https://user-images.githubusercontent.com/68846118/183392192-61478b5a-364e-4adb-84d6-c2b5aa5d688b.mp4), [video_lipmovement_w_audio_2.mp4](https://user-images.githubusercontent.com/68846118/183392279-9e798f37-7706-409f-9482-9e9553bca5d8.mp4). We also added another one with a real driving video; see [video_w_real_audio.mp4](https://user-images.githubusercontent.com/68846118/183391781-8679531a-c3ec-4fc3-aee7-d2568429600a.mp4).
> > >
> > >
> > > For 2), the experiments and analysis in the paper, supplemental materials and rebuttal have demonstrated the advantages of our method in preservation of identity, geometry, and stableness of texture in the generated facial animation videos. Our understanding is that these properties are highly important for realistic video avatars. To further verify the superiority of our method in these regards, we added a quantitative evaluation of 3D consistency and comparison among different methods (see details in the other new response). We hope this new quantitative evaluation, together with the original visual comparisons, can convince the reviewer of the advantages and potentials of our new method, which might have been overlooked by the reviewer. As stated before, we do admit that there are several limitations (such as precise facial expression and lip movement control) which prevent our method from being immediately applied in real applications. We will further discuss these limitations as the reviewer suggested. We are also willing to change our title to “towards Realistic Avatars” or “AniFaceGAN: Animatable 3D-aware Face Image Generation”. We are actually more optimistic than the reviewer and believe our method can generate face videos usable for certain real scenarios after we address the limitations.

---

### Official Review · Reviewer_jYf9 · 2022-07-18

**Rating:** 5
**Confidence:** 5
**Soundness:** 3 good
**Presentation:** 3 good
**Contribution:** 3 good

**Summary:**

This paper presented a method for learning an animatable 3D-aware generative model for face images. The method is built on top of the 3D-aware generative model GRAM and introduces 3DMM-guided imitation learning to disentangle the latent space corresponding to identity and expression variations. The method is trained on an image dataset and compared with a few previous baselines, DiscoFaceGAN, CONIFG, etc. The method is quantitatively evaluated with FID and KID, where the performance is comparable to but generally not better than previous approaches. The method also performs qualitative comparisons on mostly randomly sampled images where the authors showed better view consistency than previous work.

**Questions:**

Please discussion aforementioned questions in the rebuttal.

**Limitations:**

I believe there need to be more discussions around the limitations of the method in terms of technical details and applications, as mentioned before. The negative societal impact is adequately discussed in my opinion.

**Strengths And Weaknesses:**

The paper has the following strengths:

+ The topic is highly interesting. Controlling the trendy 3D-aware generators is of significant interest in the research communities of machine learning, computer vision, and graphics. Achieving high-quality control of 3D-aware generators is also desirable for many applications.

+ The idea is interesting and has its novelty. DiscoFaceGAN has leveraged imitation learning to learn to sample 2D faces from a 3DMM fashioned latent space. This work adopted a similar idea to the 3D-aware GANs and introduced methods to guide/regularize the 3D representations.

+ Results are encouraging. Thanks to 3D-aware GANs, the proposed method resulted in higher quality view consistency, which is generally not easy to achieve with a 2D GAN architecture.

I'm generally satisfied with the technical components and the novelty of this paper. My main concerns are regarding the quality of results and the evaluation, detailed as follows:

- In achieving facial animation, the paper works well around the mouth region but seems to fail at the eye region: the video-driven animation cannot animate motions around the eyes, such as blinks. This indicates no blinking eyes have been learned in either the 3D deformation modeling or the appearance modeling, or a combination of them. One can blame that the training data has few closed-eye images, but would data augmentation of closed-eye images resolve this problem in the 3D GAN scenario? Could 3DMM guidance be sufficient to model eye animation?

- The DiscoFaceGRAM is a baseline proposed in this method but not well explained. How does it different from the proposed approach? It looks like the DiscoFaceGRAM generated images have artifacts unseen from either DiscoFaceGAN or GRAM, Where are they coming from? Is it due to the architectural design or the training process?

- Quantitative results seem to be not better than previous approaches.

- Almost all comparisons are done with random samples across different approaches, which is not easy to interpret. The best way might be to project images into the latent space and compare the results of editing so there is a common reference. On the other hand, the video results showed some projection-then-edit animations where we can see more artifacts? How were these images/videos produced? It looks like the approach might have limitations dealing with reconstruction/protection compared to 2D StyleGANs. Perhaps more discussion is needed here too.

---

> ### Author Response · Authors · 2022-08-02
> **Response to Reviewer jYf9**
>
> We thank the reviewer for the valuable comments and suggestions. Our response can be found below.
>
> - **Eye blinking.**
> We agree with the reviewer that since we train on FFHQ where there’s almost no closed eyes, the trained model cannot generate eye motion such as blink. During training, the model almost never received input of 3DMM expression coefficients expressing closed eye; nor did it see real images with blinking/closed eyes. However, eye blinking is similar to mouth movement, both of which have some 3D content (eyeballs and oral cavity, respectively) being shown or hidden by other facial components in the front (eyelids and mouth, respectively). Therefore, we do believe this is a data issue and that adding closed-eye images will resolve this issue. Unfortunately, we were unable to collect enough image samples and retrain our model during this rebuttal period, and we will resolve this issue in the future work.
>
> - **The DiscoFaceGRAM baseline.** DiscoFaceGRAM is briefly described in Line 197-198: “*… a baseline method we call DiscoFaceGRAM, which naively combines GRAM with DiscoFaceGAN’s imitative-contrastive learning scheme*”. In DiscoFaceGAN, the GAN model is a black box to its imitative-contrastive learning scheme and thus can be replaced by any other GAN models. In DiscoFaceGRAM, we simply substituted the original StyleGAN in DiscoFaceGAN with GRAM used in AniFaceGAN while keeping other parts unchanged, and trained this so-called DiscoFaceGRAM model using its original framework. The purpose of developing such a baseline is to validate the contribution of the 3D deformation modelling and 3D-level imitation scheme we newly designed in this work. In DiscoFaceGAN and DiscoFaceGRAM, the generation process models expression change with appearance (not deformation), and all the losses are imposed on the generated 2D images. We conjecture that such strong 2D-level losses may introduce some hurdles for 3D-aware GAN training, which might have led to its lower image quality with more visual artifacts for DiscoFaceGRAM. We can further explore the reason for its lower image quality and add more discussions into the revised paper, but it should be clear that without 3D deformation modeling, such a baseline cannot guarantee exact appearance consistency when generating different expressions of the same identity.
>
> - **Quantitative results.** For image quality evaluation (FID&KID in Table 1), our result is slightly worse than GRAM, which is expected since vanilla adversarial training focuses only on image quality while we introduce controllability with additional training losses. The quality of our method is also moderately lower than DiscoFaceGAN, which is also reasonable since our backbone GRAM, as a 3D-aware GAN, still have a quality gap to traditional 2D GANs based on the powerful StyleGAN architecture. With the quick advancement of 3D-aware GAN research, we believe the gap will be narrowed quickly when applying our method to new backbones. Compared to other methods and baselines (CONFIG and DiscoFaceGRAM) that has the same goal with ours, our method has shown significantly better quantitative results.
>
> - **Visual comparison of GAN and projection-then-edit results.** We agree that visual comparison of different GAN models is difficult. Due to the random generation nature of GAN, the previous works and ours can only show different samples randomly generated by different methods. Projecting a common reference image into the latent spaces and comparing the editing results might be a possible method for comparison, but this method also has some potential limitations. Perhaps the most prominent one is the fairness of latent space embedding. Developing high-fidelity latent space embedding is still an active research topic even for well-established GAN structures like 2D StyleGAN. For 3D-aware GAN that is still in its infancy, faithful image embedding is still an open problem, and most likely will also be structure-dependent in the end. Our embedding method, which is described in Section C of the supplementary document, is a simple one without careful tweaking (e.g., we did not optimize the identity and expression coefficients; nor did we optimize the network weights). This should be the main reason for the more artifacts on the projection-then-edit results mentioned by the reviewer. We can add more discussions per the reviewer’s suggestion and explore better image embedding methods in our future work.

---

### Meta-Review · Area_Chair_mewy · 2022-08-31

**Recommendation:** Accept
**Confidence:** Certain

**Metareview:**

The paper addresses an interesting topic and advances the state of the art. The authors have responded sufficiently to the criticisms of the reviewers including the one reviewer that was recommending rejection. The authors are encouraged to incorporate the clarifications and additional results in the final version.

**Award:**

No

---

### Decision · Program_Chairs · 2022-09-14

Accept